# CLAWMACHINE: LEARNING TO FETCH VISUAL TOKENS FOR REFERENTIAL COMPREHENSION

**Tianren Ma** [*]
School of EECE
University of Chinese Academy of Sciences
matianren18@mails.ucas.ac.cn

**Lingxi Xie**
Huawei Inc.
198808xc@gmail.com

**Yunjie Tian**
School of EECE
University of Chinese Academy of Sciences
tianyunjie19@mails.ucas.ac.cn

**Boyu Yang**
Jiutian Team
China Mobile Research Institute
yangboyu@chinamobile.com

**Qixiang Ye** [†]
School of EECE
University of Chinese Academy of Sciences
qxye@ucas.ac.cn

## ABSTRACT

Aligning vision and language concepts at a finer level remains an essential topic of multimodal large language models (MLLMs), particularly for tasks such as referring and grounding. Existing methods, such as *proxy encoding* and *geometry encoding* genres, incorporate additional syntax to encode spatial information, imposing extra burdens when communicating between language and vision modules. In this study, we propose ClawMachine, offering a new methodology that explicitly notates each entity using **token collectives**—groups of visual tokens that collaboratively represent higher-level semantics. A hybrid perception mechanism is also explored to perceive and understand scenes from both discrete and continuous spaces. Our method unifies the prompt and answer of visual referential tasks without using additional syntax. By leveraging a joint vision-language vocabulary, ClawMachine integrates referring and grounding in an auto-regressive manner, demonstrating great potential with scaled-up pre-training data. Experiments show that ClawMachine achieves superior performance on scene-level and referential understanding tasks with higher efficiency. It also exhibits the potential to integrate multi-source information for complex visual reasoning, which is beyond the capability of many MLLMs. Our code is available at https://github.com/martian422/ClawMachine

## 1 INTRODUCTION

Large language models (LLMs) (Devlin et al., 2018; Brown et al., 2020; OpenAI, 2023; Gao et al., 2023; Chiang et al., 2023) have opened a new era of AI. To further unleash its potential, the researchers proposed MLLMs (Alayrac et al., 2022; Li et al., 2022; Liu et al., 2023b; Li et al., 2023a) for visual understanding and investigated the multimodal dialogue task to unify visual perception tasks. Recently, these tasks have been upgraded from image-level captioning or question answering to instance-level referring and grounding (Liu et al., 2023b; Peng et al., 2023; You et al., 2023; Ma et al., 2024), urging the MLLMs to align vision and language at a finer (*e.g.*, region or instance) level. These referential tasks require the MLLMs to understand users' intention to describe referred visual entities and predict the position information of queried objects.

---

[*]Work done during an internship at CMRI.
[†]Corresponding author.

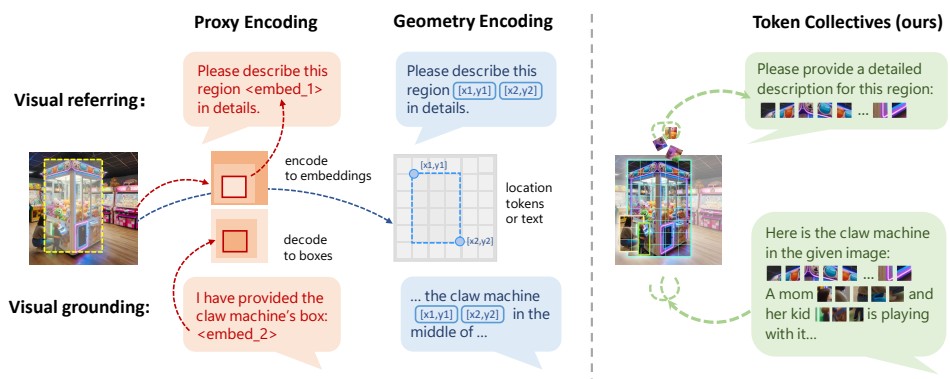

Figure 1: A conceptual comparison between existing MLLMs and our ClawMachine in notating an object in the image. ClawMachine does not use extra syntax, but directly embeds visual tokens to the natural language, supporting fine-level visual understanding in a native mechanism.

Existing methods for referential dialogues can be conceptually categorized into two genres: *proxy encoding* and *geometry encoding*. We summarize these works in Table 1 for comparison. Please refer to Figure 1 for a conceptual demonstration. The *proxy encoding* methods introduce proxy tokens as intermediates, which are processed by extra vision modules (*e.g.*, RoIAlign (Zhang et al., 2023; Ma et al., 2024), RegionCLIP (Chen et al., 2023a), SAM (Lai et al., 2023; Zhang et al., 2024; Rasheed et al., 2023), and GroundingDINO (Tian et al., 2024)) to identify objects. For these methods, MLLMs' visual grounding and referring abilities were trained separately, which brings difficulty to the joint optimization, and imposes constraints on their generalization capacity. The *geometry encoding* methods adopt an end-to-end solution by bounding objects with coordinate-like attributives. Early geometry-encoding MLLMs (Peng et al., 2023; Wang et al., 2023) report concerns about drawbacks in language ability, as they introduced exotic location tokens to align with, and the precision of grid-like data annotation was limited. Representative geometry-encoding MLLMs used plain text for location expression, but require large-scale grounded data and detailed visual information during alignment training (Chen et al., 2023c;b; Bai et al., 2023). Despite the referential understanding ability, we argue that both types of approach require additional syntax for fine-level vision-language alignment. These MLLMs' reliance on extensive instruction-tuning results in complex and burdensome training procedures, and imposes significant constraints on their generalization capacity. Please refer to Appendix A.5 for more discussion.

In this study, we propose **ClawMachine**[1], an MLLM with simple-yet-effective design to achieve referential understanding without extra syntax. ClawMachine is equipped with a new referential method that notates visual entities explicitly using **token collectives**—groups of visual tokens that collaboratively represent higher-level semantics. This avoids introducing proxy tokens or coordinates, which facilitates not only performance but also efficiency. Besides, ClawMachine introduces a **hybrid perception** mechanism, which simultaneously utilizes continuous and discrete visual information to facilitate the training procedure.

We conducted extensive experiments to evaluate ClawMachine's multimodal understanding ability. Through a designed dual-training that fits our unified format, we feed the model to outperform current models that consume much more than us, while having less hallucination on referred objects. When scaling up the pre-training data using GRIT-20M (Peng et al., 2023), the model gains larger improvement with minimal instruction tuning. Without additional codec, it demonstrates the ability to describe and ground multiple objects within one inference, and shows potential on more complex referential understanding tasks. This study reveals that, with proper design and adequate pre-training, pure auto-regressive models can outperform those with large modules and/or using heavy referential instruction-tuning. It is even one of the fastest MLLMs performing top-tier referential dialogues (see Table 16 for details). We hope this study can equip MLLMs with a native ability and shed light on unifying multiple modalities.

---

[1]The name 'ClawMachine' comes from an analogy that our model fetches the visual tokens for interpretation, just like a claw machine that grabs prizes and returns them to the player.

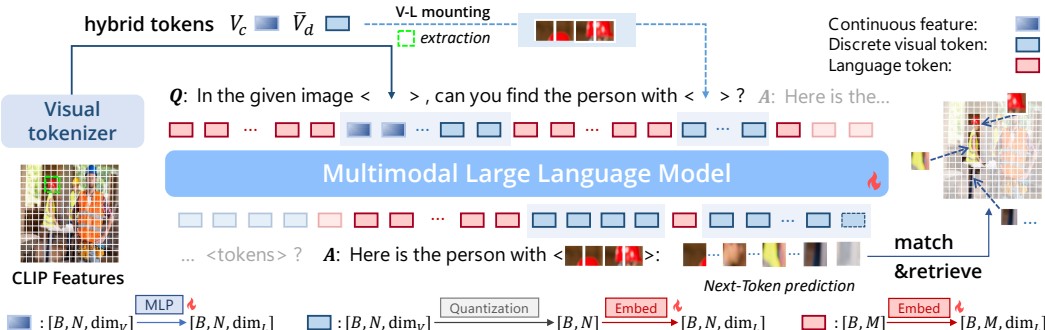

Figure 2: Framework of ClawMachine. When an image (or a region) is referred to, the corresponding visual tokens are directly embedded to the natural language. ClawMachine performs next-token prediction, and the output visual tokens are projected back to the image for grounding. $B$ represents batchsize, while $dim_V$ and $dim_L$ denotes the dimension of visual and language embeddings. *Embed* denotes LLM's embedding layer, and is demonstrated separately for intuitive explanation.

## 2 METHODOLOGY

### 2.1 MODEL ARCHITECTURE

Recent MLLMs mostly adopt architectures inspired by LLaVA (Liu et al., 2023a) to comprehend visual information. These models use a multi-layer perceptron to project image features encoded by CLIP directly onto the hidden states (embedding space) of an LLM as visual prompts. This approach enables the model to understand fine-grained visual information encoded by CLIP after training, but experience difficulty to directly generate visual tokens. To address this limitation, recent studies (Jin et al., 2023; Team, 2024) have expanded the model's vocabulary by incorporating quantized visual features as discrete tokens during training. Our approach takes advantage of both mechanisms, which we term **hybrid perception**, aiming to equip the model with more flexible visual generation capabilities while maintaining high perceptual accuracy.

To formulate the referential understanding tasks with unified notations, we utilize quantized patch features (visual tokens) alongside the text tokens in MLLM's vocabulary and use visual token collectives to notate entities within the images. The overall architecture consists of four components, namely (1) a multimodal tokenizer with hybrid perception, (2) a vision-language mounting operation, (3) a multimodal large language model for auto-regressive learning, and (4) region sampler to cluster the generated visual tokens.

**Multimodal tokenizer with hybrid perception.** For the language part, given a sentence $S$, we use LLaMA-2's tokenizer (Touvron et al., 2023) to convert it to a language token sequence $L = \{l_i\}_{i=1}^M$. For the vision part, given an image $I \in \mathbb{R}^{H \times W \times C}$, the EVA-CLIP (Sun et al., 2023b) encoder is employed to extract $N = (W/P) \times (H/P)$ features from non-overlapping patches with patch size $P$. We formulate two passes for the visual signal: the first one provides detailed visual information for scene understanding and object recognition, with the patch features projected into language model's embedding space using a 2-layer MLP, resulting in $V_c = \{v_i\}_{i=1}^N$. The second transforms the features into discrete index, where the patch features are quantized into discrete codes using LaVIT's Vector-Quantization layer (Jin et al., 2023), and result in $\bar{V}_d = \{\bar{v}_i\}_{i=1}^N$. Both $V_c$ and $\bar{V}_d$ will be used as the visual prompt for the MLLM. Please refer to Figure 2.

**Vision-language (V-L) mounting.** The key element to visual referring and grounding is the representation of referential regions. We propose to use a mounting operation to express it with token collectives: We map the bounding box as a rectangle to the image lattice and fetch the visual tokens in $\bar{V}_d$ with their center coordinates located within the rectangle. These tokens are sorted according to their center coordinates with raster order to form the token collectives, and then inserted to the dialogue correspondingly. The mounting operation can also support other types of referential regions, *e.g.*, a free-form mask.

Table 1: A detailed comparison with previous works. Grounded data refers to the instruction-tuning data for referential tasks. [G] for *geometry-encoding*, [P] for *proxy-encoding*. *: SAM used for segmenting scenes and mask decoding. † : DDETR used for object discovery.

| Method | Scene-Level Visual Perception | | Region-Level Representation | Grounded Data |
|---|---|---|---|---|
| | Encoder | Image Tokens | | |
| GPT4RoI (Zhang et al., 2023) | CLIP-H | 256 | [P] RoI Encoder | - |
| Florence-2 (Xiao et al., 2023) | DaViT-L | – | [G] Discrete bins | 5B+ |
| Shikra (Chen et al., 2023c) | CLIP-L/14 | 256 | [G] Text Coordinates | 1M+ |
| Ferret (You et al., 2023) | CLIP-L/14+ | 576 | [P] Resampler, [G] Discrete bins | 1M+ |
| Qwen-VL (Bai et al., 2023) | OpenCLIP-G (1.9B) | 1024→256 | [G] Text Coordinates | 20M+ |
| miniGPT-v2 (Chen et al., 2023b) | EVA@448px (1.0B) | 1024→256 | [G] Text Coordinates | 20M+ |
| Groundhog (Zhang et al., 2024) | CLIP/14+ & DINOv2 | 576+256 | [P] Mask Extractor, SAM* | 2.5M |
| GLaMM (Rasheed et al., 2023) | CLIP-H/14 (0.6B) | 256 | [P] RoI Encoder, SAM* | 6M+ |
| Groma (Ma et al., 2024) | DINOv2@448px | 1024→256 | [P] RoI Encoder, DDETR† | 20M+ |
| ClawMachine (ours) | EVA-G/14 (1.0B) | 256×2 | Visual Token Collectives | 700K |

**Large language model for auto-regressive generation.** With the above mechanism for tokenization and mounting, the learning objectives of both visual referring and grounding are unified in an auto-regressive form, so that the MLLM can learn more efficiently by predicting tokens using a simple classification loss. In practice, we initialize the LLM from LaVIT-2-7B (Jin et al., 2023) (a model based on LLaMA-2-7B). We provide a detailed discussion in Section 3.4 about the initialization and comparison with common LLaMAs. $L$ and $\bar{V}_d$ get vectorized using LLM's embedding layer, while $V_c$ is projected into the same language embedding space. Therefore, both $V_c$ and $\bar{V}_d$ function as the visual prompt for the LLM. This hybrid perception mechanism has been evaluated to be effective for multimodal understanding with unified vocabulary design.

**Region sampler for visual grounding.** The model is trained to interpret and generate visual tokens. During inference, a simple Gaussian Mixture Model can suggest grounding boxes based on the generated token collectives. Since the default linguistic sampling strategy, which decodes logits into tokens, is not optimal for visual tokens, we propose an efficient region sampler that incorporates object discovery priors. Details of this method can be found in Appendix A.6. This decoding strategy uses the out-of-the-box region proposer, ensuring maximum utilization of the generated tokens without causing information leakage or dispersion during training. In Section 3.1, we will discuss the intermediate metrics and grounding performance in detail.

## 2.2 DATA PREPARATION

Data preparation plays a crucial role in multimodal model research. Since the use of grounded data is often unrestricted in current MLLMs, evaluating data efficiency and the effectiveness of design can be challenging. During the training process, we made the following initial attempts.

The first model only utilize the object annotations from VG, and the RefCOCO series (including Ref-COCO, RefCOCO+, RefCOCOg; RefCOCO/+/g for short). We denote this model as **ClawMachine-7B**. The second model inherits exactly the same design but is pre-trained on 10 times larger materials including GRIT-20M (Peng et al., 2023), a highly diverse and enriched text-image dataset with phrase-level annotations. We denote this model as **ClawMachineX-7B**. The detailed data composition is summarized in Appendix. A.2.

In the pre-training stage, scene-level captioning data like LLaVA-Pretrain and ShareGPT-4V, region-level captioning data from RefCOCOg and VG are utilized. For ClawMachineX, we transform GRIT-20M to an image dataset with interleaved region-text captions. The format is described as follows:

```
Hybrid image tokens: <boi><feats_c><eoi> <boi><feats_d><eoi>
A transformed caption example: <ref> Mother <boi><ref_tokens><eoi>
and <ref> the little daughter <boi><ref_tokens><eoi> in
<ref> hats <boi><ref_tokens><eoi> and...
```

In the instruction-tuning stage, we mainly train ClawMachine to answer two kinds of referential questions, *i.e.*, visual referring and grounding, and show that it generalizes to more complex scenarios. The input and output of visual referring are curated into the following format, as

```
User:  In the given image <boi><feats_c><eoi>
<boi><feats_d><eoi>, please provide a detailed description
for this <ref> region <boi><ref_tokens><eoi>.
Assistant:  <A detailed description of the region>.
```

When the input is loaded, `<boi><feats_c><eoi>` will be substituted with $V_c$ and `<boi><feats_d><eoi>` substituted with $\bar{V}_d$. The `<ref_tokens>` will be substituted with the token collectives extracted by V-L mounting. A trigger token `<ref>` is placed before the entity, notifying the MLLM that visual tokens will follow. Two special tokens, `<boi>` and `<eoi>`, are used to wrap these visual tokens. Similarly, the input and output of visual grounding are curated into the following format:

```
User:  In the given image <boi><feats_c><eoi>
<boi><feats_d><eoi>, can you find [object]?
Assistant:  Here is <ref> [object] <boi><ref_tokens><eoi>.
```

where `[object]` can be substituted with any text description of the object. With the unified next-token prediction objective, we can combine the curated datasets for referring and grounding and train a single model for both tasks. We call it the *dual* training data and will show in Section 3.4 that it benefits the model's performance. Refer to Appendix. A.2 for more data samples.

## 2.3 IMPLEMENTATION

The training process is partitioned into two stages, supervised by a next-token prediction loss. Please refer to Appendix A.3 for training configuration.

**Stage 1: Alignment pre-training.** We guide the model in performing basic captioning task at this stage. The hybrid visual sequences are used. For ClawMachine-7B, a mixture of scene-level and region-level captioning data is used. At the end of this stage, we obtain an MLLM that can generate captions based on features of various scopes. For ClawMachineX-7B, extra 15M non-instruction-tuning training data is used, and the MLLM can generate interleaved image-text token sequences. The MLP and entire language model is trained while other components are frozen. Note that the MLP projector is priorly initialized. See Section 3.4.

**Stage 2: Instruction-tuning.** We get the model accustomed to general VQA and referential dialogues in this stage. We collect and curate visual instruction-tuning data from various sources for this stage. The model's referring and grounding ability is trained simultaneously with the *dual* curated data, which is verified effective to improve the model's grounding performance. The MLP and entire language model is trained while other components are frozen. All the experimental results are tested after this stage without task-specific fine-tuning.

## 3 EXPERIMENTS

### 3.1 REFERENTIAL COMPREHENSION

**Visual Referring.** We first evaluate ClawMachine on the visual referring task to assess its ability of region-level understanding. The prompt for visual referring has the form of `Please provide a detailed description for this <ref> region <ref_tokens>`, where `<ref_tokens>` is replaced by the visual tokens within the target region as explained in the V-L mounting part. Table 2 presents our results on two established region captioning benchmarks, RefCOCOg (Kazemzadeh et al., 2014) and Visual Genome (Krishna et al., 2016). With scaled-up pre-training data, our model can better capture the token collective's semantics.

Table 2: **Results on the visual referring task.** ClawMachine demonstrates state-of-the-art performance among recent MLLMs.

| Model | RefCOCOg | | Visual Genome | |
|---|---|---|---|---|
| | METEOR | CIDEr | METEOR | CIDEr |
| GRiT (Wu et al., 2022) | 15.2 | 71.6 | 17.1 | 142 |
| Kosmos-2 (Peng et al., 2023) | 14.1 | 62.3 | - | - |
| GPT4RoI (Zhang et al., 2023) | - | - | 17.6 | 146.8 |
| Shikra-7B (Chen et al., 2023c) | 15.2 | 72.7 | - | - |
| GLaMM-7B (Rasheed et al., 2023) | 16.2 | 105.0 | 19.0 | 163.9 |
| Osprey-7B (Yuan et al., 2024) | 16.6 | 108.3 | - | - |
| Groma-7B (Ma et al., 2024) | 16.8 | 107.3 | 19.0 | 158.4 |
| **ClawMachine-7B (ours)** | 17.1 | 115.4 | 19.3 | 168.9 |
| **ClawMachineX-7B (ours)** | **17.4** | **118.4** | **19.5** | **169.7** |

**Visual Grounding.** Next, we study visual grounding, *a.k.a.* referring expression comprehension (REC), the counterpart task to visual referring that requires the model to identify the location of an object with language descriptions. By utilizing discrete visual tokens for image encoding, ClawMachine can understand visual content like reading a paragraph. Consequently, cross-domain grounding is transformed into a token retrieval task in the joint vision-language vocabulary. We use `In the given image, can you find [object] ?` as the question. The results are shown in Table 3.

Table 3: **Results on the visual grounding (REC) task.** We report accuracy with the IoU threshold 0.5. *: Groma interprets and selects pre-detected sequences instead of generating grounding tokens. [†]: OFA, UniTAB and Florence-2 are generalist foundation models.

| Model | RefCOCO | | | RefCOCO+ | | | RefCOCOg | | Data |
|---|---|---|---|---|---|---|---|---|---|
| | val | test-A | test-B | val | test-A | test-B | val | test | |
| OFA-L[†] (Wang et al., 2022a) | 79.9 | 83.7 | 76.4 | 68.3 | 76.0 | 61.8 | 67.6 | 67.6 | 6M+ |
| UniTAB[†] (Yang et al., 2022) | 88.6 | 91.1 | 83.8 | 81.0 | 85.4 | 71.6 | 84.6 | 84.7 | 1.5M+ |
| Florence-2[†] (Xiao et al., 2023) | 93.4 | 95.3 | 92.0 | 88.3 | 92.9 | 83.6 | 91.2 | 91.7 | 5B+ |
| Shikra-7B (Chen et al., 2023c) | 87.0 | 90.6 | 80.2 | 81.6 | 87.4 | 72.1 | 82.3 | 82.2 | 1M+ |
| MiniGPT-v2 (Chen et al., 2023b) | 88.7 | 91.6 | 85.3 | 80.0 | 85.1 | 74.5 | 84.4 | 84.7 | 20M+ |
| Qwen-VL-7B (Bai et al., 2023) | 88.5 | 92.3 | 84.5 | 82.8 | 88.6 | 76.8 | 85.9 | 86.3 | 20M+ |
| Ferret-7B (You et al., 2023) | 87.5 | 91.3 | 82.5 | 80.8 | 87.4 | 73.1 | 83.9 | 84.8 | 1M+ |
| Groma-7B* (Ma et al., 2024) | 89.5 | 92.1 | 86.2 | 83.9 | 88.9 | 78.0 | 86.4 | 87.0 | 20M+ |
| **ClawMachine-7B (ours)** | 88.6 | 92.1 | 84.3 | 83.2 | 88.2 | 76.9 | 85.7 | 86.3 | 700K |
| **ClawMachineX-7B (ours)** | **89.7** | **92.5** | **86.9** | **84.4** | **88.9** | **78.0** | **86.7** | **87.1** | 16M |

We use the region sampler described in Section 2.1 to convert model's output visual tokens into bounding boxes required by visual grounding benchmarks. In this model, we have also used the *dual* data composition policy (*i.e.*, by including the training data curated for visual referring) to improve the performance. The comparison of ClawMachine against existing MLLMs for visual grounding is shown in Table 3. ClawMachine reports state-of-the art performance among the MLLMs that utilize extensive training data or specifically designed architectures for visual grounding.

Additionally, with region-text interleaved GRIT-20M data during pre-training, ClawMachineX is capable of locating multiple objects and provide a description within one inference, which most of current MLLMs cannot perform. Following Groma (Ma et al., 2024), we conduct experiments with its LVIS-Ground benchmark, which focuses on on testing the model's ability to locate multiple, diverse, and variably-sized objects. See Appendix A.7 for details. The results are summarized in Table 4. ClawMachineX demonstrates superior ability with its native interleaved token generation ability, and surpasses existing models especially on small objects. The lack of small objects in popular datasets like RefCOCO/+/g is one important reason, besides, as discussed in Section 3.2, ClawMachine can recognize visual concepts in quantized tokens, which provides extra convenience for fetching small objects that occupy few patches in the image.

Table 4: **Results on LVIS-Ground benchmark.** ClawMachineX demonstrates superior capability for grounding multiple objects at once.

| Model | AR@s | AR@m | AR@l | AR |
|---|---|---|---|---|
| Shikra-7B (Chen et al., 2023c) | 0.1 | 3.1 | 18.5 | 4.9 |
| MiniGPT-v2 (Chen et al., 2023b) | 0.3 | 8.0 | 41.1 | 11.4 |
| Ferret-7B (You et al., 2023) | 1.6 | 16.7 | 51.1 | 16.8 |
| Groma-7B (Ma et al., 2024) | 8.7 | 35.6 | 64.3 | 28.8 |
| **ClawMachineX-7B (ours)** | **20.5** | **43.1** | **69.2** | **36.7** |

Table 5: Evaluating intermediate results using the three metrics in equation 1 on RefCOCOg-val.

| Training data | precision | recall | IoU | REC |
|---|---|---|---|---|
| Visual Genome | 29.1 | 39.5 | 36.8 | 74.7 |
| RefCOCO/+/g | 26.2 | 42.3 | 37.0 | 70.1 |
| + Visual Genome | 33.1 | 45.5 | 52.8 | 85.7 |
| + GRIT-20M | 42.1 | 51.2 | 54.8 | 86.7 |
| GRIT-20M | 30.4 | 42.7 | 44.5 | 80.3 |

**Quantitative evaluation of the token collectives.** As the intermediate result of visual grounding, the quality of generated tokens heavily impacts the accuracy of grounding. Initially, the MLLM does not always generate complete visual tokens that cover the entire target. We considering several token sets, where $\mathcal{G}_{\text{image}}$, $\mathcal{G}_{\text{pred}}$, and $\mathcal{G}_{\text{gt}}$ denote the set of whole-image tokens, predicted tokens, and ground-truth tokens, respectively. Note that the ground truth is extracted from the bounding box, which may contain some background tokens. We then define four metrics for precision, recall, and IoU:

$$
\begin{aligned}
\text{precision} &= |\mathcal{G}_{\text{pred}} \cap \mathcal{G}_{\text{gt}}| \, / \, |\mathcal{G}_{\text{pred}}| \\
\text{recall} &= |\mathcal{G}_{\text{pred}} \cap \mathcal{G}_{\text{gt}}| \, / \, |\mathcal{G}_{\text{gt}}| \\
\text{IoU} &= |\mathcal{G}_{\text{pred}} \cap \mathcal{G}_{\text{gt}}| \, / \, (|\mathcal{G}_{\text{pred}} \cap \mathcal{G}_{\text{image}}| + |\mathcal{G}_{\text{gt}}| - |\mathcal{G}_{\text{pred}} \cap \mathcal{G}_{\text{gt}}|)
\end{aligned}
\tag{1}
$$

The results of different ClawMachine variants are summarized in Table 5, revealing several important insights. (1) The precision of retrieved tokens improves as the amount and diversity of training data increases. (2) The recall of retrieved tokens is higher when the training data is from the same domain (RefCOCO/+/g), but data from other domains (such as Visual Genome) helps to fill gaps. This suggests that increasing the diversity of training data is an effective strategy. After scaling with GRIT-20M, the quality of token collectives shows significant improvement. The visualization results in Figure 8 further support this observation. As GRIT-20M supports the model in contrastively learning object representations and their features (the associated image tokens in this study), it leads to improvements in IoU and precision. To further investigate the model's performance with controlled object size and IoU thresholds, we also made extended evaluations in Appendix A.4.

## 3.2 FINDINGS

**ClawMachine learns visual concepts in quantized tokens.** Figure 3 visualizes visual grounding examples. One can see that the retrieved tokens are strongly correlated to the query. Delving into the output, we find that ClawMachine learns to connect visual words in the joint vocabulary with linguistic concepts, *e.g.*, `tennis ball` is correlated with the `13635`-th visual word and `person` (in small scale) is correlated with the `14781`-th visual word. Particularly, This has also been validated in LVIS-Ground evaluations on small objects in Table 4. The basic understanding of individual visual tokens can be extended to token collectives, which collaboratively represent higher-level semantics. This makes it much easier for the MLLM to perform visual grounding: it only needs to fetch the tokens that are most related to `[object]` from the image.

**Token collectives alleviate referential hallucination.** Referential dialogues place higher demands on a model's ability to recognize and localize objects. However, current referential visual instructions can make MLLMs prone to hallucination, as many scenes and details are repeated across different auto-generated datasets. To address this, we conducted additional experiments to evaluate the model's referential hallucination. Inspired by POPE (Li et al., 2023b), we curated a Ref-Hal test based on GQA (Hudson & Manning, 2019), where the model is asked to verify answers based on referring objects. Details of the curation and testing process are provided in Appendix A.8. The model can only give correct answers if it truly understands the user's referential intent. The results, summarized in Table 7, show that ClawMachine exhibits significantly fewer hallucinations in object references under the same non-tuning setting. Proxy-encoding models underperformed, despite some showing strong results in POPE, which we attribute to their customized design for current tasks. Geometry-encoding model Qwen-VL showed better overall performance, while Shikra lagged behind, likely due to its multiple passes over datasets during training.

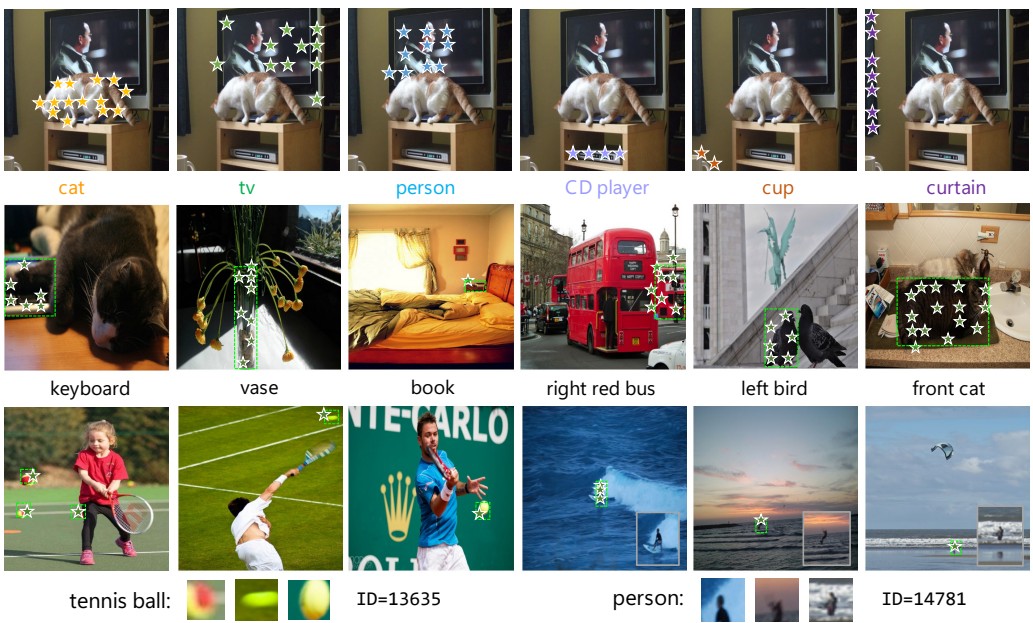

Figure 3: ClawMachine generates visual tokens, projects them to the image lattice (denoted by stars), and predicts the grounded box (denoted by rectangles). The top row shows the ability to ground different objects within one image. The bottom row shows visual tokens with the same ID.

Table 6: **POPE adversarial result for object hallucination.** Tokens indicate the image tokens used for each image. *: Groundhog uses DINOv2 feature fusion, and traversed much more data.

| Method | Tokens | Accuracy | Precision | Recall | F1 score | Yes(%) |
|---|---|---|---|---|---|---|
| LLaVA-7B (Liu et al., 2023b) | 256 | 50.77 | 50.39 | 99.87 | 66.98 | 99.10 |
| Shikra-7B (Chen et al., 2023c) | 256 | 83.10 | 85.6 | 79.60 | 82.49 | 46.50 |
| Ferret-13B (You et al., 2023) | 576 | 82.36 | 83.60 | 80.53 | 82.00 | 48.18 |
| Groundhog-7B* (Zhang et al., 2024) | 576+256 | 86.33 | 85.93 | 86.63 | 86.28 | 49.60 |
| ClawMachine-7B (ours) | 256×2 | 85.36 | 86.39 | 82.87 | 84.59 | 48.10 |

Table 7: Results on our designed hallucination test with referred objects (Ref-Hal).

| Method | Referring Syntax | Accuracy | Precision | Recall | F1 score | Yes(%) |
|---|---|---|---|---|---|---|
| Shikra-7B (Chen et al., 2023c) | Text Coordinates | 47.3 | 48.3 | 76.0 | 59.1 | 78.7 |
| Qwen-VL-7B (Bai et al., 2023) | Text Coordinates | 61.5 | 58.3 | 81.0 | 67.8 | 69.5 |
| Ferret-7B (You et al., 2023) | RoI Features | 56.8 | 54.9 | 76.0 | 63.8 | 69.2 |
| GPT4RoI-7B (Zhang et al., 2023) | RoI Features | 44.0 | 45.7 | 63.4 | 53.1 | 69.4 |
| Groma-7B (Ma et al., 2024) | RoI Features | 53.9 | 52.8 | 72.8 | 61.2 | 68.9 |
| ClawMachine-7B (ours) | Token Collectives | 75.8 | 72.8 | 82.4 | 77.3 | 56.6 |

**Complex visual comprehension.** ClawMachine benefits from two-fold advantages: (i) it has a native ability to perform visual referring and grounding so that *a question and its answer can contain both image and text elements*, and (ii) it builds a clear relationship between visual and linguistic tokens so that *the same concept can be delivered using either image or text*. Integrating these advantages allows it to solve complex visual reasoning tasks. Figure 4 shows examples including: *a.* grounding multiple objects at once. *b.* grounding-upon-referring, where the query to visual grounding contains image-embedded tokens, and *c.* multi-object referring segmentation, where multiple sets of retrieved tokens are converted into instance segmentation results. it also supports *d.* and *e.* free-form and *f.* region-text interleaved VQA. These are beyond the capability of current methods within one model, and shows ClawMachine's potential towards complex and flexible referential dialogues.

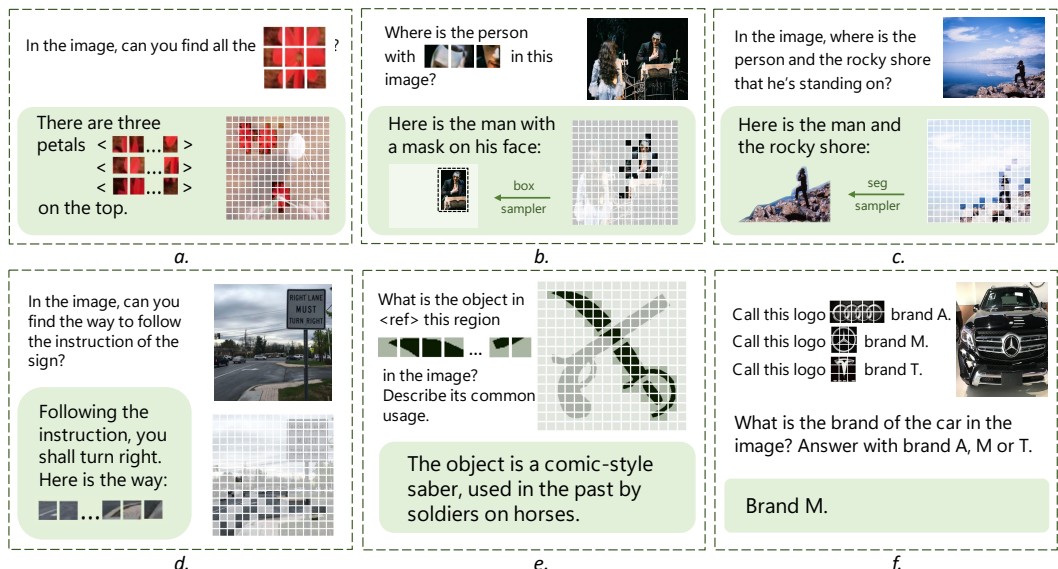

Figure 4: ClawMachine can solve complex visual reasoning tasks. See the texts for explanations.

## 3.3 SCENE-LEVEL PERFORMANCE

In addition to region-level tasks, we further evaluate ClawMachine on conversational VQA benchmark LLaVA Bench (Liu et al., 2023b), and model's performance on object hallucination benchmark POPE (Li et al., 2023b). The LLaVA Bench contains questions about conversations, detailed descriptions and complex reasoning. As shown in Table 8, ClawMachine maintains competitive image understanding and visual chatting abilities. Without high-resolution visual encoders and task-specific fintuning, ClawMachine gains the highest average score comparing with other MLLMs doing VQA tasks. We show the adversarial set's results on POPE as a complement to the Ref-Hal results. ClawMachine also demonstrates competitive results without data or feature augmentation.

## 3.4 DISCUSSION AND ABLATIVE STUDY

**Initialization.** In exploring recent MLLMs with a joint vision-language vocabulary Jin et al. (2023); Team (2024), we base our model on LaVIT for several reasons: as an early endeavor towards unified vision-language generation model, LaVIT provides us with an MLLM with extended vision vocabulary and corresponding VQ model, utilizing large amount of data for text and image generation. Notably, LaVIT uses the MLP projector and standalone visual tokenizer weights for multimodal understanding, while has not been trained on COCO captioning or VQA datasets. So we excluded all these weights to restore the MLLM for a fair initialization. Prior to pre-training, we retrained the projector to facilitate an equitable comparison with other models on scene-level understanding. We initialized the projector from scratch with learning rate of $1 \times 10^{-3}$, and used the CC-SBU-558k data introduced in LLaVA for training. During this process, we disable quantization and only input continuous patch embeddings $V_c$, while keeping the tokenizer and MLLM frozen.

**Hybrid Perception.** A straightforward choice is to use either continuous or discrete image features. However, our findings reveal some interesting insights: As can be seen in Table 9, the quantization loss of discrete image tokens negatively impacts the model's visual perception. Despite this, quantization is pivotal for visual grounding, as discrete tokens are easier to retrieve in our next-token prediction framework. We conducted experiments by toggling between continuous and discrete modes. The limitations of using separate modes are overcome by using both $V_c$ and $\bar{V}_d$. We attribute this improvement to the separate optimization of the embedding states for the two signals-using MLP for $V_c$ and LLM's embedding layer for $\bar{V}_d$-which collectively enhances the adaptation of the original CLIP features. The experiments are conducted under the default setting with RefCOCO/+/g and VG data.

Table 8: Results on LLaVA-Bench (COCO).

| Method | Conv. | Desc. | Reas. | Avg. |
|---|---|---|---|---|
| LLaVA-7B (Liu et al., 2023b) | 85.4 | 68.3 | 92.1 | 81.9 |
| Kosmos-2 (Peng et al., 2023) | 71.7 | 63.4 | 74.9 | 70.0 |
| Shikra-7B (Chen et al., 2023c) | 80.6 | 70.7 | 88.1 | 79.9 |
| Ferret-7B (You et al., 2023) | 84.4 | 79.4 | **96.3** | 86.7 |
| Groma-7B (Ma et al., 2024) | 82.6 | **84.0** | 88.8 | 85.2 |
| ClawMachine-7B (ours) | **84.8** | 82.5 | 94.5 | **87.3** |

Table 9: Ablation on image feature settings.

| Stage-1 | Stage-2 | Referring | Grounding | VQAv2 |
|---|---|---|---|---|
| *conti.* | *conti.* | 16.8 | 73.1 | 78.4 |
| *conti.* | *discr.* | 16.5 | 81.1 | 77.9 |
| *discr.* | *discr.* | 14.2 | 84.3 | 72.1 |
| *discr.* | *conti.* | 14.4 | 75.8 | 72.3 |
| concat(*conti.*, *discr.*) | | 17.1 | 85.7 | 78.9 |

Table 10: Ablation on Model design.

| Encoder | Syntax | Referring | Grounding | VQAv2 |
|---|---|---|---|---|
| Continuous | text | 14.9 | 81.0 | 78.4 |
| Continuous | v. tokens | 15.8 | 79.1 | 78.4 |
| Hybrid | text | 14.9 | 81.1 | 78.6 |
| Hybrid | v. tokens | 17.1 | 88.6 | 78.9 |

Table 11: Ablation on Encoder design.

| Encoder | Precision | IoU | REC |
|---|---|---|---|
| EVA-CLIP | 33.1 | 52.8 | 85.7 |
| EVA-L/14 | 30.3 | 51.4 | 83.9 |
| CLIP-H/14 | 31.4 | 51.9 | 84.4 |
| CLIP-L/14 | 29.5 | 51.0 | 83.4 |

**Model Design Ablation.** We added more ablation studies in Table 10 under the strict same pre-training and instruction-tuning datasets, *i.e.,* same as used by ClawMachine-7B. We remove global discrete tokens in *continuous*, and did ablation experiments by replacing reference visual token collectives (v. tokens) with text coordinates. Referring results was tested on RefCOCOg using the METEOR score and grounding on RefCOCO-val.

**Encoder Ablation.** We compared model's performance with different encoders in Table 11. We pick EVA-CLIP as the image encoder for the following reasons: First, it is pre-trained with a mask image modeling procedure, which wre validated enjoying perfect semantic re-construction on visual token collectives. Even at the same size, the visual encoders with MIM pretraining scored better (compare EVA-L (EVA02) and CLIP-L). Second, the LaVIT's vanilla quantization layer (a 2-layer MLP with a indexing in the codebook), which helps to get the index of visual tokens, is tuned to tokenize EVA-CLIP features. We replace it with an ordinary ViT, and observe certain drop in grounding benchmarks. This can also be attributed to insufficient training of the quantization layer. For comparison, Shikra, which uses CLIP-L as the encoder, scored 82.3 on the REC score.

## 4 CONCLUSION

We present ClawMachine, a multimodal large language model designed to unify various fine-level referential tasks. Central to ClawMachine is the approach of annotating visual entities with corresponding tokens, eliminating the need for additional syntax. This method demonstrates the model's ability to grasp high-level semantics through token collectives. Furthermore, ClawMachine excels in complex visual reasoning by seamlessly integrating visual and language tokens within the same sequence. Our research indicates that this unified design, combined with effective pre-training, enables pure auto-regressive models to outperform those with large modules and extensive referential instruction tuning.

**Limitation and future work.** Vision data is rich in information, but quantizing visual features often results in loss of fine-grained details. We anticipate the development of more advanced encoding mechanisms that minimize information loss when embedding visual data into the joint vocabulary(Tschannen et al., 2023). Additionally, the probabilistic nature of large language models can lead to unstable and incomplete outputs for visual tokens—for instance, failing to capture the entire grounded target. We envision a unified learning procedure that integrates the LLM with the visual encoder to address this challenge effectively.

## 5 ACKNOWLEDGMENTS

The authors thank Yuan Zhang for region sampler exploration, Yuzhong Zhao for technical discussion, and the GPU resources provided by the Jiutian Team of China Mobile Research Institute. We also thank the anonymous reviewers for their helpful comments. This work was supported by National Natural Science Foundation of China (NSFC) under Grant 62225208, 62450046 and CAS Project for Young Scientists in Basic Research under Grant No.YSBR-117.

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

## A APPENDIX

We provide the related work, some additional explanation and details in the Appendix section.

### A.1 RELATED WORK

**Multimodal large language model.** The community is witnessing a trend towards unifying the vision and language modalities using multimodal large language models (MLLMs) (Alayrac et al., 2022; Li et al., 2023a; Liu et al., 2023b). Early endeavors like Flamingo adapted LLMs to visual tasks by internal cross-attention design (Alayrac et al., 2022), while BLIP-2 (Li et al., 2023a) introduced Q-former as an external block for vision-language alignment. Later works like LLaVA (Liu et al., 2023b;a) applied effective projectors for alignment pre-training, providing a ViT-MLP-LLM architecture for the community alongside the visual instruction-tuning method. Recently, a stream of works (Kondratyuk et al., 2023; Lu et al., 2023) unified more modalities like video and audio into the model's vocabulary with advanced autoencoder designs (Yu et al., 2023), paving a promising path for large multimodal models.

**Referential notation.** Referential notations are important for MLLMs to align text and image modalities at a finer level. In methods that utilized *proxy encoding* (Zhang et al., 2023; Yuan et al., 2024; Chen et al., 2023a; Rasheed et al., 2023), the model leverages the referential proxy as a signal for agents and incorporated a specially-designed module to encode region-features. Powerful foundation models like SAM (Kirillov et al., 2023) are also utilized to perform object discovery (Rasheed et al., 2023; Zhang et al., 2024). When asked to ground visual objects, the model produces an intermediate embedding supervised by detection or segmentation decoders (Lai et al., 2023; Tian et al., 2024). Another research line applies the *geometry encoding* that utilizes discrete tokens to annotate the boundary of instance and train the MLLM in an end-to-end manner, providing a unified method for the model to recognize and generate instance positions (Chen et al., 2023c; Xuan et al., 2023; Peng et al., 2023; You et al., 2023; Xiao et al., 2023).

**Tokenization of vision.** Aligning visual tokens within the language space has gained significant attention in recent research. Using CLIP features of image patches, the Emu series (Sun et al., 2023c;a) combined visual generation tasks with multimodal comprehension, while LaVIT (Jin et al., 2023) introduced an extended visual vocabulary to simplify training objectives. More recently, auto-regressive generation with directly quantized image patches (Team, 2024; Sun et al., 2024) has sparked a new trend in the community. Although these early alignment methods are computationally intensive to replicate, they provide valuable foundation models for experimentation. In this work, we aim to further strengthen the spatial correlation within the multimodal embedding space.

### A.2 DATA COMPOSITION DETAILS

We provide data usage details in this section.

During pre-training, the data is transformed into a simple concatenation of visual signals and corresponding description with no instruction-tuning style templates. For scene-level captioning, the tokens of entire image are used, while only the cropped tokens from the V-L mounting operation are employed for region-level captioning. Some images of GRIT-20M are no longer available from the Internet, and we do a simple filtration by removing images that focus on celebrities and pure text images, so the final size of available data of GRIT is about 15M.

During fine-tuning, we mix the general VQA datasets with referential dialogues. We only reserve the COCO-VQA and GQA subset of LLaVA-mix665k, as the text-related subsets are of severe hallucination and do not help with general understanding tasks (model performs better with cleaned data on VQAv2 bench). Besides classical referential instructions, we collect some useful subsets from existing and widely used datasets. Osprey makes short annotations from original RefCOCO datasets more vivid and detailed. AS-V2 provides more flexible referring and grounding conversations, which helps to interleave vision and language tokens. Specifically, scene-graph like data are utilized for describing and grounding multiple objects at one inference with an aligned output style. The claimed 700K grounded data refers to the sum-up of {RefCOCO/+/g, Visual Genome, Osprey: detailed, AS-V2: scene graph, AS-V2: conversation, and Chatterbox: CoQ}. Please see Table 12 for more information.

**Effectiveness of dual data.** We study the impact of using different instruction tuning data in Stage 2, and evaluate the model's visual grounding performance. We compare the contribution of the *plain* data (the dialogue data without region-level questions, sourced from LLaVA-mix665k (Liu et al., 2023a)) and *dual* data (the visual referring part), by only adding either of them apart from the grounding data, and find that the latter brings about 1.7% gain in REC average score (although the former is also useful in maintaining the model's VQA ability). This validates that ClawMachine can absorb the knowledge from various kinds of referential dialogues, which mainly owes to its formulation that unifies referring and grounding into the next-token prediction task.

Table 12: Data composition details.

| Training Stage | Data | Description and Usage | Image-Text Pairs |
|---|---|---|---|
| Pre-training | LLaVA-Pretrain (Liu et al., 2023b) | Scene-Level Caption | 558k |
| | ShareGPT-4V (Chen et al., 2023d) | Scene-Level Caption | 100k |
| | RefCOCO/+/g (Kazemzadeh et al., 2014) | Region-Level Caption | 287k |
| | Visual Genome (Krishna et al., 2016) | Region-Level Caption | 247k |
| | GRIT-20M (Peng et al., 2023) | Interleaved Caption (for ClawMachineX) | 15M (filtered) |
| Fine-tuning | LLaVA-mix665k (Liu et al., 2023b) | General VQA | 436k (filtered) |
| | RefCOCO/+/g (Kazemzadeh et al., 2014) | Visual Referring&Grounding | 287k |
| | Visual Genome (Krishna et al., 2016) | Visual Referring&Grounding | 247k |
| | Osprey: Detailed (Yuan et al., 2024) | Referring Details on RefCOCO | 63k |
| | AS-V2: Scene Graph (Wang et al., 2024) | Multiple Grounding on RefCOCO | 42k |
| | AS-V2: Conversation (Wang et al., 2024) | Refer&Ground convs. on RefCOCO | 22k |
| | Chatterbox: CoQ (Tian et al., 2024) | Logical Chain of Grounding | 40k |

## A.3 TRAINING DETAILS

We use the AdamW (Loshchilov & Hutter, 2019) optimizer with the cosine annealing scheduler (He et al., 2019) to adjust the learning rate. The initial learning rate is set to $2 \times 10^{-5}$ and $1 \times 10^{-5}$ for the two stages respectively with a warm-up ratio of $0.03$. The global batch size remains constant at $256$. We freeze the tokenizer and train the projector and LLM in both stages. The training is conducted on 8×NVIDIA A100 GPUs with 80GB memory. The FlashAttention-2 and DeepSpeed libraries with *zero2* are employed for efficient training. The input image size is set to $224 \times 224$ with a patch size $P = 14$, and the maximum sequence length in the MLLM is $2048$. The codebook of the VQ process is 16384, while the language tokenizer has 32000, resulting in the MLLM with 48384 vocabulary size. The training datasets are combined into a single dataloader using the V-L mounting operation. The image-text pairs are randomly selected during training and are only traversed for one epoch in each training stage. The training takes about 8 hours for each stage of ClawMachine-7B, and the pre-training of ClawMachineX-7B takes about 3.5 days. We provide a configuration list for all the components that are tuned. The tokenization layer of visual tokens is frozen throughout the process, while the corresponding hidden state is trained within LLM's embedding layer. The *Projector alignment* is stated in Section 3.4 for a fair comparison. Note that our model's training

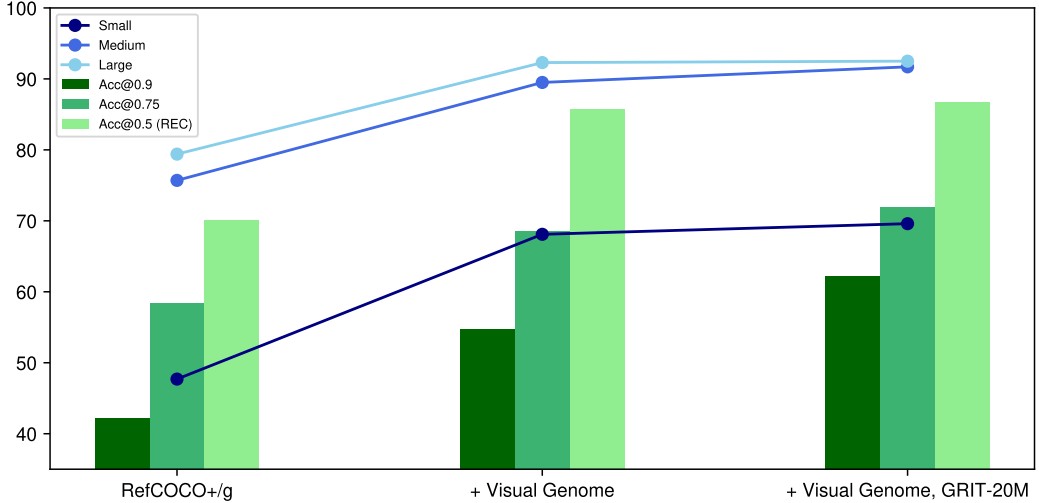

Figure 5: Model's detailed grounding performance with controlled IoU thresholds and object sizes. Note: small(23.6%): $S \in (0, 0.08)$, medium(49.1%): $S \in (0.08, 0.25)$, large(27.3%): $S \in (0.25, 1)$

strategy, which is addressed as *visual instruction tuning* (Liu et al., 2023b), is widely utilized by current state-of-the-art multimodal language models.

Table 13: A comparison of major dataset (>100k pairs) usage among popular models.

| Method | RefCOCO/+/g | VG | Flickr30K | Object365 | GRIT-20M | Others |
|---|---|---|---|---|---|---|
| Shikra (Chen et al., 2023c) | ✔ | ✔ | ✔ | ✗ | ✗ | VCR (Zellers et al., 2019) |
| GLaMM (Rasheed et al., 2023) | ✔ | ✔ | ✔ | ✗ | ✗ | SA-1B (Kirillov et al., 2023) |
| Qwen-VL (Bai et al., 2023) | ✔ | ✔ | ✗ | ✗ | ✔ | In-house Data |
| Ferret (You et al., 2023) | ✔ | ✔ | ✔ | ✔ | ✗ | LVIS (Gupta et al., 2019) |
| Groundhog (Zhang et al., 2024) | ✔ | ✔ | ✔ | ✔ | ✗ | VCR, GQA (Hudson & Manning, 2019) |
| Groma (Ma et al., 2024) | ✔ | ✔ | ✔ | ✗ | ✔ | ShareGPT-4V-PT (Chen et al., 2023d) |
| ClawMachine (ours) | ✔ | ✔ | ✗ | ✗ | X Pre-train | ShareGPT-4V (Chen et al., 2023d) |

Table 14: Training Configuration.

| Training Stage | MLP Projector | LLM | Data usage | Strategy |
|---|---|---|---|---|
| Projector alignment | Training | Frozen | CC-SBU-558k Standard LLaVA-pretrain | $lr = 1 \times 10^{-3}$ 1 epoch |
| Pre-training | Training | Training | 1.2M/16M caption data Denoted in Table 12 | $lr = 2 \times 10^{-5}$ 1 epoch |
| Fine-tuning | Training | Training | 700k instruction-tuning data Denoted in Table 12 | $lr = 1 \times 10^{-5}$ 1 epoch |

## A.4 FURTHER EVALUATION ON GROUNDING METRICS.

An initial exploration of size-based evaluation in LVIS-Ground is presented in Table 4, we conduct further evaluation here with controlled IoU thresholds to highlight the model's performance differences under incremental data settings. We calculate the *relative area S* by multiplying object's relative $w$ and $h$ (between 0 and 1). To ensure the consistency with the REC metric, the $x$ of Acc@$x$ here is calculated using the IoU of final grounding results and ground truth. All the results are evaluated on RefCOCOg-val.

As shown in figure 5, pre-training on GRIT-20M led to consistent improvements in performance across IoU scales, with particularly significant advantages at higher IoU thresholds. The model also demonstrated enhanced detection capabilities for objects of varying sizes.

## A.5 AN INTUITIVE EXPLANATION ABOUT REFERENTIAL COMPREHENSION.

For existing methods, the LLM learns to understand the reference expressed by newly introduced token or embeddings. Specifically, for *geometry-encoding* methods, the grounding ability is represented by its learning of discrete location token or numerical coordinates. This brings high demand on image's resolution and feature details for the alignment. As can been seen in the comparison Table 1, while the grounded data demand is high for pre-training, most *geometry-encoding* methods also employ higher-resolution encoders to guarantee the perception details.

For *proxy-encoding* methods, the representation of referred objects is based on vision priors like RoI embedding. However, while the embedding may be updated to align with the language space, its relationship with the original vision features is not optimized. This makes the proxy embedding task-specific and limits MLLM's generalization capacity. Moreover, the grounding embedding weaved by the hidden stated of last output tokens makes it technically hard to localize multiple objects during inference, let alone the decoder-specific design on rare foundation models like SAM and GroundingDINO. Groma tackles this issue by pre-detecting all objects in an image and guiding the model to select the correct responses, although this increases the tuning complexity and latency.

ClawMachine enhances referential tasks by integrating language and vision more deeply within an auto-regressive architecture. Using a joint vocabulary, it represents references through token collectives, facilitating a stronger fusion of language and vision in the embedding space. This approach helps align finer concepts and allows the model to learn high-level semantics across multiple visual tokens, which has been roughly encoded by CLIP yet not utilized by existing MLLMs. Additionally, this hybrid perception design promotes joint optimization, improving the overall performance in referential comprehension tasks.

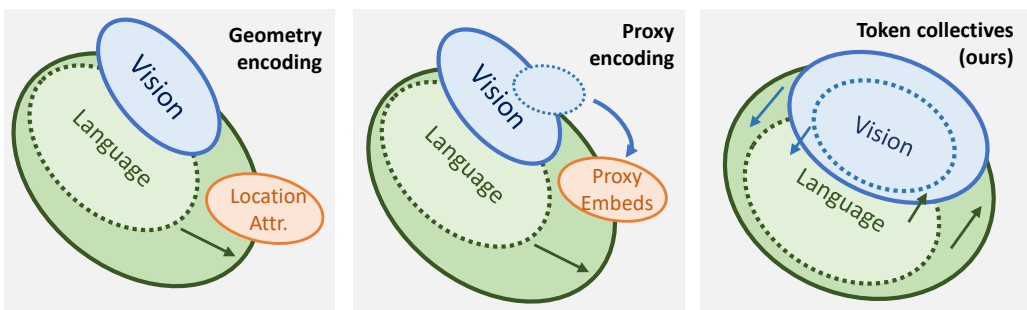

Figure 6: A conceptual illustration about referential MLLMs. See text for explanation.

## A.6 REGION SAMPLER DESIGN

We explain how our grounding method works and provide some details for decoding visual tokens into bounding boxes in this section.

**Retrieving Tokens in Image.** Let $\mathcal{G}_{\text{image}}$ represent the group of image tokens, $\mathcal{G}_{\text{pred}}$ the group of generated tokens. We get $\mathcal{G}_{\text{pred}\cap\text{image}} = \mathcal{G}_{\text{image}} \cap \mathcal{G}_{\text{pred}}$ as the primary material for further processing. For each element $v_i \in G_{pred\cap image}$, as its index in the visual tokenizer's codebook is determined, we can easily retrieve its original place with $\bar{V}_d = \{\bar{v}_i\}_{i=1}^N$. As the tokenizer has limited vocabulary and may not assign every patch of the image a different code, we firstly retrieve visual tokens that appear only once in $\mathcal{G}_{\text{image}}$. Then for $v_i$ that appears more than once in $\mathcal{G}_{\text{image}}$, we scan all of its possible origins and append the index that is nearest to the already picked tokens to minimize extra noise. After this procedure, we can get a retrieval map for model's output visual tokens, which just looks like the stars on a canvas as we illustrated in Figure 9.

**Region Sampler with Detection Priors.** As the default linguistic sampling strategy that decodes logits into tokens are not optimal for visual tokens, we propose an efficient region sampler with object discovery priors, which is similar to the modern *beam search* (Lemons et al., 2022) algorithm utilized by LLMs:

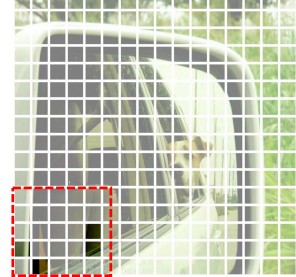

**Where is the person?**

First, a switchable region proposer is deployed for a primary object detection. For a given image, the proposer provide prediction boxes with a score higher than 0.3 by default, and we use them as the region proposals. We do not give it any clue like class or description. See Table 15 for our experiments. Hit-Rate refers to the chance that the proposer can discover the object described by the dataset with IoU > 0.5. Some annotations are quite difficult even for human in RefCOCO series, and for the rest of the objects, the discovery results are close.

Figure 7: A figure that is difficult for proposers. As no proposal is valid, ClawMachine will output the convex box for the decoded tokens.

Second, with the logits of generated token collectives, we conduct a fuzzy search after the the retrieval. For tokens that do not have explicit match in the image grid, their logits are decoded correspondingly if the top-3 index fall into the region proposals' lattice.

Only the proposals that already have decoded tokens will be considered in the fuzzy search. If no tokens have fell into the proposals' lattice with overlap > 0.1, we skip the nomination process and use a simple convex box for the decoded tokens as the result which is effective for small or hindered objects.

Finally, for each proposed box, we initialize a 2D Gaussian distribution based on it. Suppose the gaussian distribution's variance shall be inversely proportional to box's size, while the covariance matrix of the distribution follows the shape of the rectangle. We use these pre-defined distributions to construct a simple Gaussian Mixture Model (GMM), and predict the points' intention. This will result in a prediction label for each point that decides which distribution it belongs. After using the prediction results and points' density to score each box, we nominate the box with highest scores as model's prediction results. For multiple object detection, as each generated token collectives is bounded with their object name in "`<ref> object <boi> <ref_tokens> <eoi>`", this process is executed sequentially with no overlap.

**A Primary Latency Test.** The efficiency of MLLMs has often been overlooked in recent referential studies. However, we believe it remains a crucial indicator of a model's capabilities. Geometry-encoding methods typically employ an end-to-end training procedure, where processing detailed visual tokens—often at higher resolutions—accounts for most of the inference time. In contrast, proxy-encoding methods approach referential dialogues in an agent-like manner, utilizing LLMs to weave embeddings while relying on large foundation models as peripheral modules for downstream decoding. This coupling introduces higher latency during both training and inference.

In our approach, the proposer used in our region sampler also contributes to processing time during inference. So we conducted an average latency comparison with the released versions of these models (note that Groundhog is not yet released), as shown in Table 16. An esimation of FLOPs(Floating Point Operations) is also conducted to evaluate model's efficiency, as shown in Table 17. With the switchable proposer design, ClawMachine offers users flexibility in choosing between precision and efficiency. The latency was tested and averaged using the default evaluation scripts provided by the developers (where available), or we used similar questions to those posed to ClawMachine.

## A.7 TESTING ON LVIS-GROUND

The LVIS-Ground benchmark introduced by Groma (Ma et al., 2024) focuses on grounding multiple objects in the image. They randomly sample at most 5 images for each object category from the LVIS (Gupta et al., 2019) validation set to construct LVIS-Ground. The AS-MANY-Protocol is followed for evaluation: this protocol selects the top-k predicted boxes (where k is the number of ground-truth boxes) and measures recall over all ground-truth boxes. For example, if there are 3 out of 5 ground-truth boxes hit by the top-5 predicted boxes, the recall is 60%. More details can be seen in the main manuscript paper.

Table 15: Ablation on Region Sampler's design choice. The REC score is tested on RefCOCO-val with ClawMachineX. Hit-Rate refers to the chance that the proposer can discover the object described by the dataset with IoU > 0.5. NOTE: The previous best is 89.5 reported by Groma (ECCV'24), which adopts a large DDETR-like detector pre-trained on SA-1B to extract all the objects in a scene, taking more than **500ms** for each round detection.

| Proposer | Hit-Rate (%) | Latency (ms) | REC score |
|---|---|---|---|
| Co-DETR(Swin-L) (Zong et al., 2022) | 92.5 | 217 | 89.7 |
| Co-DETR(R-50) (Zong et al., 2022) | 92.1 | 165 | 89.1 |
| YOLOX-X (Ge et al., 2021) | 89.5 | 17 | 88.7 |

Table 16: A inference latency comparison with other models. The precision is set to bfloat16. Note that our model does not incorporate any detection modules during training. The inference time of ClawMachine and its X version keeps the same. *: with Co-DETR or YOLOX-X as the proposer.

| Model | Image tokens | Modules | Referring (ms) | Grounding (ms) |
|---|---|---|---|---|
| Ferret-7B | 576 | Spatial Resampler | 378 | 431 |
| GLaMM-7B | 256 | SAM | 449 | 639 |
| Groma-7B | 576+256 | DDETR | 733 | 757 |
| ClawMachine-7B | 256×2 | Region Sampler | 309 | 577 / 377* |

Table 17: A FLOPs estimation of different methods. The precision is set to bfloat16. The computation cost of ClawMachine and its X version keeps the same. *: with Co-DETR or YOLOX-X as the proposer.

| Model | Vision Encoding | Language Decoding | additional modules | Total (TFLOPs) |
|---|---|---|---|---|
| Kosmos-2 (1B) | 0.1 | 0.1 (~300 tokens) | – | ~0.2 |
| Shikra-7B | 0.1 | 0.6 (~300 tokens) | – | ~0.7 |
| Ferret-7B | 0.15 | 1.4 (~650 tokens) | 0.03 (Resampler) | ~1.6 |
| GLaMM-7B | 0.2 | 1.0 (~400 tokens) | 1.0 (SAM) | ~2.2 |
| Groma-7B | 0.4 | 0.8 (~350 tokens) | 0.8T (DDETR) | ~2.0 |
| ClawMachine-7B | 0.27 | 1.1 (~550 tokens) | 0.4 / 0.1 (Region Sampler) | ~1.8 / 1.5 |

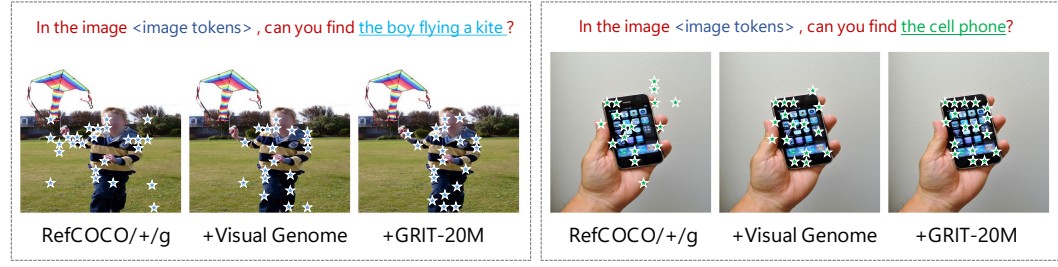

Figure 8: The diversity of training data improves the accuracy of retrieved visual tokens.

In each inference of ClawMachine, it outputs a scene-graph like sentence describing all the instances in the image with grounding token collectives. For multiple objects with a same category, it knows to use `<boi>` * `<eoi>` to wrap up each objects' predicted tokens, just like Kosmos-2 (Peng et al., 2023) with GRIT uses the similar syntax to ground multiple objects. We first map all the output grounding tokens into boxes using the region sampler. In the image, for each category in the ground-truth, we calculate whether ClawMachine mentions it, and the corresponding boxes will be used to calculate the recall. The average recall is calculated over 10 IoU thresholds (ranging from 0.5 to 0.95) as the primary metric on LVIS-Ground.

## A.8    REF-HAL: EVALUATING REFERENTIAL HALLUCINATION

Inspired by POPE (Li et al., 2023b), we curate a simple Ref-Hal test dataset based on GQA (Hudson & Manning, 2019), asking the model to make verification based on referring objects. Here is the details about Ref-Hal:

Table 18: A simplified demonstration of GQA composition. The data types that do not exist in GQA is denoted with gray cells. Only the types with checkmarks ✓ are sampled.

| | Compare | Logical | Verify | Query | Choose |
|---|---|---|---|---|---|
| Category | | | | | |
| Attribute | ✓ | ✗ | ✓ | ✗ | ✗ |
| Relation | | | ✓ | ✗ | ✗ |
| Object | | ✗ | ✗ | ✗ | ✗ |
| Global | | | | | |

Our primary intention for Ref-Hal is to evaluate model's hallucination on referred objects, while avoid overlapping with existing benchmarks on complex referential understanding abilities. The GQA serves as an ideal start point, as it focuses on forming strong bonds among objects in VG images. We make an summary in Table 18. As shown, the composition of GQA can be roughly categorized with two dimensions: semantic (vertical) and operation (horizontal). The *logical*, *query*, and *choose* operation is firstly excluded for the following reasons: *logical* relies on knowledge rather than recognition of objects, and overlaps with *verify* on question styles; *query* has no referred objects in the question; *choose* focus on the detailed attributes of the single object, which can be evaluated with general visual referring benchmarks. *verify-object* is excluded, too, as it asks the model whether something is in the image, much like the POPE for hallucination tests. The remaining three sets have the following format:

*Compare-Attribute:* "Is the <obj_1> SAME <attr.> <obj_2>?" to verify the attributes of two referred objects. Examples: Do the mouse and the curtain have the same color? Is the bag made of the same material as the door frame?

*Verify-Attribute:* "Is the <obj> <attr.> adj.?" to verify certain attribute of the referred object. Example: Does the shirt has green color? Are the bleachers that are not long made of wood?

*Verify-Relation:* "Is the <obj_1> <relation> <obj_2>?" to verify the relationship of two referred objects. Example: Is the person wearing a hat? Is the car behind a motorcycle?

We extract 221, 334, and 445 Q&A pairs from GQA's testdev-balanced set correspondingly (same as the original ratio of these three types in the testdev-balanced set), and $yes : no = 1 : 1$. We utilize GPT-4V to replace one object in half of the questions with another object that appears in the same image, assuring that the answer is reversed (the replacement choice is random like POPE random set, but not replacing it with something that is not in the image, which makes any reference invalid). Then, we transform the object notations in the question with `object` in the referential style. After this modification, the question "`Is the car behind a motorcycle?`" is transformed into "`Is the object-1[R₁] behind a object-2[R₂]?`". Where $[R_n]$ is the referential notation. As GQA do not explicitly annotate the objects with coordinates, this substitution used the original ground-truth annotation of VG to ensure correspondence and precision.

With the curated Ref-Hal benchmark, we test model's hallucination on referential objects. The results are shown in Table 7. The model needs to recognize the referred object in the question before answering: *e.g.*, "`object[0.113,0.224,0.355,0.998]`" for geometry-encoding models, "`object[RoI embedding]`" for proxy-encoding models, and "`object[token collectives]`" for ClawMachine. The model only needs to answer yes or no, and the evaluation result is summarized in Table 7. For a fair comparison, ClawMachine is not trained on any subsets of this benchmark.

## A.9    MORE VISUALIZATION EXAMPLES

We provide more visualization examples of ClawMachine's output in this part. As the resulted points serve as a *task-agnostic supervision* intermediate for downstream tasks, we also use it as a prompt

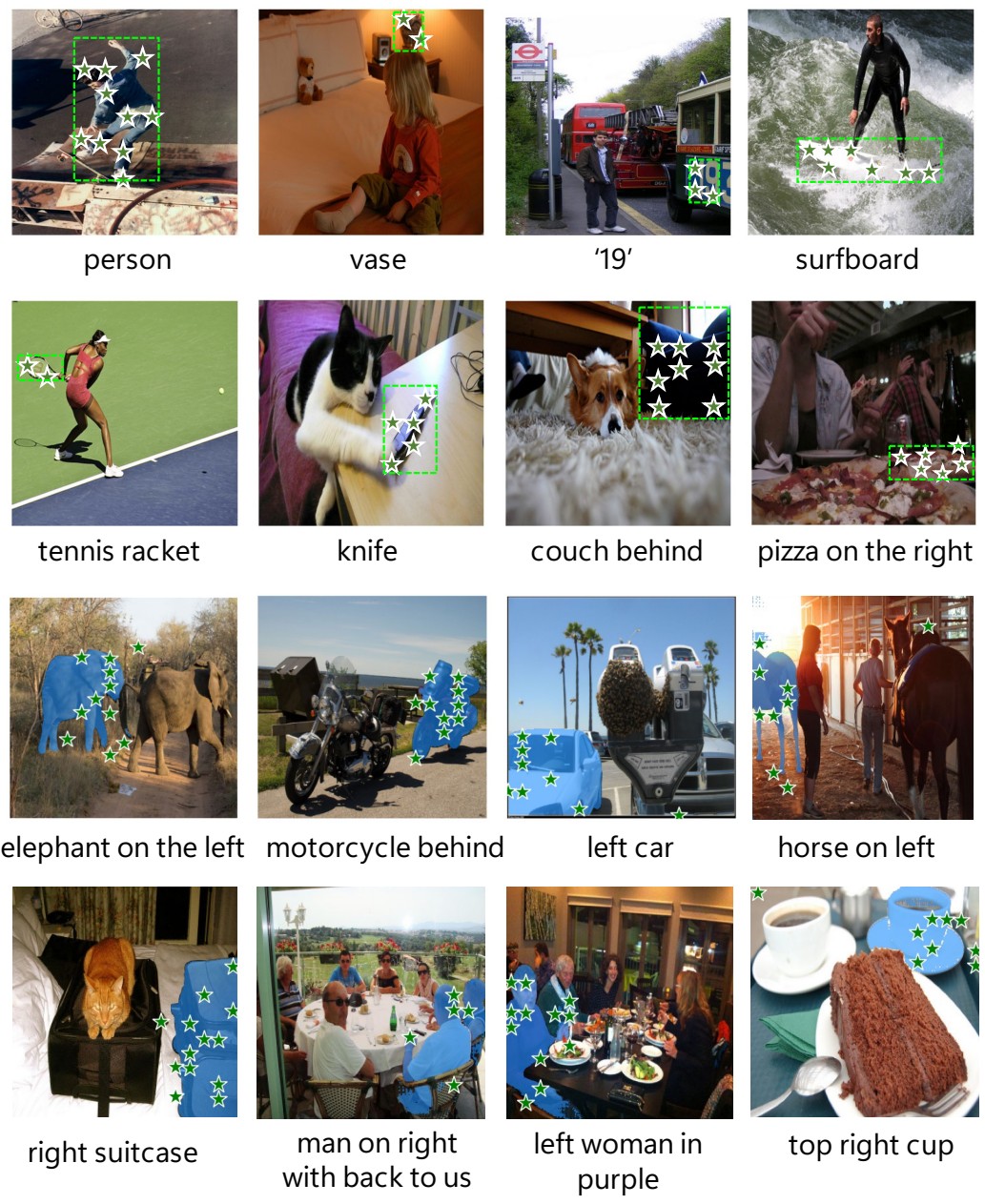

Figure 9: More visualization examples of ClawMachine. The bottom two rows show some segmentation results we get from SAM using points' clustered center as the prompt.

for powerful pre-trained segmentation models (Kirillov et al., 2023). The points' convex closure can be regarded as a semantic segmentation with rough granularity, which is finely post-processed by segmentation specialist models. We show some examples in Figure 9. However, using multiple points as a weak supervision is of limited support from SAM and other segmentation models, related segmenting works like LOST (Siméoni et al., 2021) and TokenCut (Wang et al., 2022b) are still under development. We hope to address this problem with a sampler with finer granularity and free-form support in the future.

