# OpenReview forum: "ClawMachine: Learning to Fetch Visual Tokens for Referential Comprehension"
_ICLR.cc/2025/Conference — ICLR 2025 Poster_

### Official Review · Reviewer_3BfC · 2024-11-03

**Soundness:** 3
**Presentation:** 3
**Contribution:** 3
**Rating:** 6
**Confidence:** 4

**Summary:**

This paper proposes to use token collectives for representing visual concepts in MLLMs, which avoids learning proxy or geometric tokens. As a result, the visual referring and grounding tasks can be prompted and answered in a unified form. The extensive experiments demonstrate the effectiveness of the proposed method.

**Strengths:**

1. The idea of utilizing token collectives for object referring and grounding is innovative and may inspire future research on integrating multiple modalities.
2. The experimental results show the effectiveness of the proposed method.
3. The paper is well-written and easy to follow.

**Weaknesses:**

1. The input image is encoded to 16*16=256 patches, leading to a large number of tokens for referring or grounding an object. For grounding an object, geometry-encoding methods and proxy-encoding methods only one or a few tokens, while ClawMachine needs to output hundreds of tokens (for a large object), which largely increase the inference time cost. However, Table 13 does not appear to reflect this discrepancy in grounding time. Could you provide more details on how the inference time is measured and if there are any techniques used to maintain efficiency despite the potential for increased token output?
2. The Region Sampler relies on a pre-trained detector to obtain grounding results, which may hinder overall model performance if the detector fails to identify the target object. This reliance on detection could limit scalability compared to one-stage methods such as Ferret and GLaMM.
3. As stated in the Limitation section, the vector quantization of visual features results in a loss of fine-grained details, as well as a larger vocabulary usage than previous methods. This also poses challenges on aligning linguistic and visual tokens to some extent.

**Questions:**

Please refer to the weaknesses section.

---

### Official Review · Reviewer_E6or · 2024-11-04

**Soundness:** 3
**Presentation:** 2
**Contribution:** 3
**Rating:** 8
**Confidence:** 3

**Summary:**

The paper proposes ClawMachine, a new method to train MLLMs for referring and grounding. The method allows a fine-grained alignment of vision and language in large transformers. It introduces “token collective”, which are groups of visual tokens representing regions of images. ClawMachine uses text-vision interleaved data and simple next token prediction to train MLLMs. ClawMachine can then perform several tasks, such as referring and grounding, together with other complex tasks necessitating a fine vision-language understanding.

**Strengths:**

* ClawMachine does not require additional syntax to embed the localisation.

* ClawMachine can be trained with the simple next token prediction task and instruction tuning.

* ClawMachine has strong experimental results in referring and grounding.

* The paper claims that models and data will be made available.

**Weaknesses:**

* The section `Methodology` is very succinct and lacks some motivations:
    * Using both continuous and discrete tokens (only motivated in experiments)
    * Training details are tedious as ClawMachine leverages multiple existing elements. It would make it easier to follow to have a detailed table indicating the parts that are used from previous works / and which are fine-tuned or frozen.
    * The visual token generation is not clear to me, is the supervision the patch index, or the vocabulary index?

* ClawMachine is a supervised method. ClawMachine relies on datasets such as RefCOCOg etc. which are supervised datasets. The question remains of how scalable this method is.

* The paper does not include the recent Florence-2 [1]; which can perform both referring and grounding. It also has much fewer parameters.

**Questions:**

* What are the performances of the model after pre-training but without instruction tuning?

* Table 6 shows that adding the GRIT-20M dataset as pre-training data leads to large improvements for the proposed precision / recall / IoU. However this does not lead to such a gap in REC performances. Have the authors interpretation on this?

* In future works the authors mention not relying on quantized tokens. Would the type of approach described in GIVT [2] be an interesting way of doing so?

[1] Xiao, Bin, et al. "Florence-2: Advancing a unified representation for a variety of vision tasks." Proceedings of the IEEE/CVF Conference on Computer Vision and Pattern Recognition. 2024.

[2] Tschannen, Michael, Cian Eastwood, and Fabian Mentzer. "Givt: Generative infinite-vocabulary transformers." European Conference on Computer Vision. Springer, Cham, 2025.

---

### Official Review · Reviewer_gC1c · 2024-11-05

**Soundness:** 3
**Presentation:** 2
**Contribution:** 2
**Rating:** 5
**Confidence:** 3

**Summary:**

This paper proposes a novel approach for aligning vision and language concepts in multimodal large language models. Unlike other works in the literature that use additional syntax to encode spatial information, this approach introduces a new strategy for representing entities using token collectives—groups of visual tokens that represent entities at a semantic level

**Strengths:**

1. The idea of representing entities using the introduced concept of a 'token collective' is novel and interesting, as it eliminates the need to train a separate object detector in 'proxy encoding' approaches, allowing for end-to-end joint optimization. Additionally, it removes the difficulty in aligning language with exotic location tokens in the other 'geometry encoding' methods.

2. The proposed solution is simple yet effective. Extensive experiments across several datasets and settings are conducted to show the approach's effectiveness. The paper also provides adequate ablation studies to analyze the model's behavior. The qualitative results presented in Figures 3 and 7 provide insights and illustrate the proposed approach's ability to generate relevant visual tokens.

**Weaknesses:**

While the method itself is not very complex, the presentation in the paper is difficult to follow and understand the details. The two most important figures (Figures 1 and 2) are not referenced anywhere in the main text. The core section of the paper, Section 3.1, contains very limited information. Although additional explanations are provided in Appendix A.4, relying solely on textual explanations makes some aspects ambiguous.

Please also see further concerns in the question section below.

**Questions:**

- The paper uses an out-of-the-box object detector, as mentioned in line 205 and Table 12. Since this is also a pre-trained external model, what is the advantage of this in comparison with other detector modules in 'proxy encoding' and 'geometry encoding' methods?

- As stated in line 821: '...We do not give it any clue like class or description...' Is it correct that the region sampler can propose an arbitrary number of boxes (as long as the confidence score is greater than 0.3), and that all tokens within these boxes are extracted to form the 'token collective' set?

- In Table 1, the proposed method uses another image encoder, EVA-G/14 (1.0), compared to previous methods. Since the proposed strategy for extracting 'Token Collectives' does not depend on the choice of image encoder, how would the method's performance be affected if it used the same encoder as in previous works?

- In Table 6, why does training on RefCOCO/+ and RefCOCOg lead to better recall and IoU in the Token Collectives generation step when compared with Visual Genome, yet result in a lower REC score? Additionally, what are the results when training only on GRIT-20M? (rather than on RefCOCO+GRIT-20M)

- In line 188, the paper states that tokens are sorted with primary sorting from top to bottom and secondary sorting from left to right. Will the results be impacted if random shuffling or another sorting strategy is applied?

- Is there any experiment conducted for "free-form masks" rather than bounding boxes, as mentioned in line 190?

---

> ### Author Response · Authors · 2024-11-24
>
> Dear Reviewer,
>
> We sincerely appreciate the time and effort you have dedicated to reviewing our work. As the author-reviewer discussion period draws to a close, we kindly seek your feedback on whether our responses have adequately addressed your concerns.  If you have any additional suggestions or comments, please do not hesitate to share them with us. We are more than willing to engage further and make any necessary improvements!
>
> Thanks again for your thoughtful insights and valuable contributions.

---

> ### Author Response · Authors · 2024-11-27
> **We look forward to receiving your feedback.**
>
> Dear Reviewer,
>
> There is one day remaining before the cutoff for revising the PDF. We look forward to receiving your feedback on our responses and welcome any further discussion in the coming days.
>
> Thank you again for your time and consideration!

---

### Official Review · Reviewer_8JFN · 2024-11-11

**Soundness:** 3
**Presentation:** 3
**Contribution:** 3
**Rating:** 6
**Confidence:** 5

**Summary:**

The paper proposed a hybrid way to incorporate continuous features and discrete tokens in MLLM for referring and grounding. Different from using coordinate location to represent regions, this work introduced discrete image tokens in both input and output. A systemic workflow is proposed to accommodate the change. In the experimental result, the proposed model can outperform previous works in multiple tasks and shows good qualitative analysis.

**Strengths:**

1. The idea of token collectives enables referring and grounding in MLLM in a hybrid way, unifying both continuous region features and discrete region tokens.

2. Instead of generating coordinates as in previous works, this paper proposed to generate discrete tokens for corresponding regions, and then do a patch-based matching and retrieval to ground back to the region in the image. Also, a delicately designed resampler is introduced to ensure the accuracy of grounding boxes.

3. The performance is impressive, outperforming many existing works in a wide range of tasks.

**Weaknesses:**

1. It's a bit unclear whether the good performance comes from data or model design. I.e., the proposed model design needs to be ablated and compared with other baselines in the same training and dataset setup as used in this paper. For example, there are a few necessary ablations I could think of: replacing the referring LaVIT token with discrete coordinates; replacing the grounding LaVIT token to be predicted with discrete coordinates, remove the LaVIT token for the global image. Please make sure the other settings are the same (using the same pre-training and instruction-tuning datasets as in the paper, the same training stages and steps, same backbone).

2. Efficiency and Complexity: The proposed method requires two image encoders: LaVIT for discrete token generation and Eva-CLIP for continuous token. Also in decoding, the proposed method requires a detector to provide prior in the Region Sampler. Essentially, other existing works don't require that many additional components. A detailed comparison in terms of efficiency analysis (total model parameter, FLOPs, memory) with and without detector/LaVIT is needed.

3. An advantage of hybrid representation is about free-form referring and grounding(segmentation). However, this paper doesn't benchmark that quantitatively.

**Questions:**

Please see the weaknesses. Will consider revising the review if the author(s) can solve my concern in rebuttal.


===========================

After reading the rebuttal of the author(s), I'd like to raise my rating as my concerns are alleviated.

---

### Meta-Review · Area_Chair_ZZpr · 2024-12-18

**Metareview:**

This paper proposed a novel approach for referring and grounding in MLLM. Concretely, the tokenized image patch is directly served as input and output target for the MLLM. This novel design explores a new direction for adding the position information to the MLLM. The proposed approach achieves good performance on the benchmarks.

The reviewers support this paper for:
1. novel architecture design
2. strong performance
3. simplified architecture / method

The reviewers argue this paper for:
1. whether data / model contributes to the final perfomrance.
2. presentation of this paper.
3. efficiency of the proposed approach.
4. results for the free-form referring and grounding.

Based on the discussion, majority of the reviewers agree that the pointed out weaknesses have been addressed. Only reviewer gC1c didn't respond to the rebuttal. Based on the rebuttal, I would rule that the concerns have been addressed.

Based on the support, I vote this paper for accept.

**Additional Comments On Reviewer Discussion:**

Before the rebuttal, the reviewers have concerns about the presentation of the paper / cleanness. The revision of this paper makes the figure more easier to be understood.

The reviewers also concerns the major contribution of the improved performance. The author also included additional results to clarify this concern.

Finally, for the free-form referring and grounding, the author provides additional results for that during rebuttal.

Based on the above, I think the author addressed those point nicely.

---

### Decision · Program_Chairs · 2025-01-22

Accept (Poster)